# The Interplay of Stress, Inflammation, and Metabolic Factors in the Course of Parkinson’s Disease

**DOI:** 10.3390/ijms252212409

**Published:** 2024-11-19

**Authors:** Tal Ben Shaul, Dan Frenkel, Tanya Gurevich

**Affiliations:** 1Movement Disorders Center, Neurological Institute, Tel Aviv Medical Center, Tel Aviv 6423906, Israel; bensha92@gmail.com; 2School of Medicine, Tel Aviv University, Tel Aviv 6997801, Israel; 3Department of Neurobiology, School of Neurobiology, Biochemistry and Biophysics, Tel Aviv University, Ramat-Aviv, Tel Aviv 6997801, Israel; 4Sagol School of Neuroscience, Tel Aviv University, Tel Aviv 6997801, Israel

**Keywords:** Parkinson’s disease, neurodegeneration, neuroinflammation, physiological stress, psychological stress, diabetes, insulin signaling and resistance, metabolic factors, dopaminergic degeneration

## Abstract

Parkinson’s disease (PD) is a prevalent neurodegenerative condition for which there are symptomatic treatments but no disease-modifying therapies (DMTs). Extensive research over the years has highlighted the need for a multi-target DMT approach in PD that recognizes the various risk factors and their intricate interplay in contributing to PD-related neurodegeneration. Widespread risk factors, such as emotional stress and metabolic factors, have increasingly become focal points of exploration. Our review aims to summarize interactions between emotional stress and selected key players in metabolism, such as insulin, as potential mechanisms underlying neurodegeneration in PD.

## 1. Introduction

Parkinson’s disease (PD) is a prevalent neurodegenerative disorder that was initially viewed as a purely motor disorder associated with Lewy bodies and the loss of dopaminergic neurons in the substantia nigra. PD is now recognized as a much more complex neurological condition with affective, cognitive, autonomic, and other non-motor manifestations. The pathology of PD involves extensive regions of the nervous system, various neurotransmitters, and numerous protein aggregates. The cause of PD remains unknown. It appears to result from a complex interplay of genetic and environmental factors that influence numerous critical cellular processes [1].

There are pharmacological and non-pharmacological methods to control the motor aspects of PD, but considerable efforts have not yet yielded disease-modifying therapy to date. It has been speculated that new targets for intervention may emerge as the pathophysiology of PD becomes further understood [2].

Neurodegeneration in PD has been associated with impaired insulin signaling, insulin resistance, mitochondrial dysfunction, oxidative stress, impaired glucose utilization, metabolism, and inflammation [3,4,5].

Stress is characterized as a state in which external factors create adverse conditions for the individual, originating from various sources such as physical threats, biological conditions, or psychological states [6]. The association between stress and neurodegenerative disorders, including PD, has been hypothesized since the 1980s [7,8].

In this review, we aim to highlight the potential multidirectional interactions between metabolic disorders, such as insulin resistance and type 2 diabetes mellitus (T2DM), inflammatory factors, physiological/psychological stress, and the risk and manifestation of PD (Figure 1).

## 2. Physiological Stress and PD

Physiological stress is a state in which external factors create adverse conditions for the individual stemming from various sources, such as physical threats, biological conditions, or psychological states [6]. This stressor triggers a reaction known as the stress response, commonly referred to as the “fight or flight” response, which occurs through the release of stress hormones, such as cortisol, epinephrine, glucagon, and growth hormone. This response involves three pathways: (1) the hypothalamic–pituitary–adrenal (HPA) axis and its hormonal components, including corticotropin-releasing hormone, adrenocorticotropic hormone, and glucocorticoids and mineralocorticoids; (2) the renin-angiotensin system (RAS), with angiotensin II as a potent vasoconstrictor and aldosterone; and (3) the autonomic nervous system (ANS) via the sympathetic or parasympathetic nervous system [9].

A growing number of publications support the long-held view that acute or chronic stress is a major contributor to the development of PD [10,11,12]. The onset of neurodegenerative disease symptoms can be triggered by stress resulting from dysfunctions that developed years earlier [12,13,14]. In describing his first patient, James Parkinson in 1817 emphasized that the symptoms occurred after an emotionally stressful event, a pattern that was later reported by others as well [15]. More than 100 years ago, Dr. William Richard Gowers wrote that prolonged anxiety and emotional shock are “the most common antecedents of Parkinson’s disease” and advised his patients to refrain from “all causes of mental strain and of physical exhaustion” [16]. Furthermore, severe psychological stressors, such as those experienced by soldiers in World War I, the Holocaust, and as prisoners of war, have been linked to PD [7,17,18,19]. Until the end of the 20th century, there was comparatively little in the scientific literature about the role of stress in the etiology of neuropathologies. However, with advanced insights into the pathophysiology of neurodegeneration, this link is now being systematically investigated in PD as well as in Alzheimer’s disease [20].

Prolonged stress was shown to trigger oxidative stress, as evidenced by increases in protein and lipid peroxidation and DNA damage [21]. Neurons in the brain are constantly exposed to reactive oxygen species (ROS) and reactive nitrogen species due to endogenous or exogenous exposure to oxidative stress [22]. Oxidation of catecholamines leads to the formation of quinones, which are capable of triggering lipid peroxidation and membrane disruption [23] as well as impairing cell viability [22,24]. These characteristics make oxidative stress a factor in the cascade leading to dopamine agonist cell degeneration involving mitochondrial dysfunction, excitotoxicity, nitric oxide toxicity, and inflammation [24]. The GI system is another factor attributed to the connection between stress and PD. “Stressful eating” can increase oxidative stress resulting in the production of pro-inflammatory cytokines, lipid peroxidation, and the disruption of insulin signaling while leading to mitochondrial and DNA damage and further amplifying ROS production, establishing a detrimental feedback loop [25]. A study conducted with a rotenone PD mouse model demonstrated that stress led to gut barrier dysfunction, marked by increased permeability and inflammation and higher levels of endotoxins in the bloodstream. Stress further altered the gut microbiome by decreasing beneficial bacteria and increasing pro-inflammatory bacteria. Mice exposed to stress demonstrated increased neuroinflammation, particularly in the substantia nigra; higher *α*-synuclein levels; more severe motor deficits; and neuronal degeneration [26]. Alterations in the gut microbiome can also result in the increased production of microbial lipopolysaccharide (LPS), triggering additional inflammatory reactions [27].

Animal studies support the possible link between physiological stress, dopaminergic degeneration, and PD [28,29,30,31,32,33,34]; however, they are beyond the scope of this review.

## 3. Psychological Stress

It has been established that PD is associated with psychological stress, which is characterized as a psychological response to a breakdown in stress adaptation that includes symptoms such as debilitating fatigue, and it is significantly linked to the risk of developing PD [35]. Depression alone increases the risk of PD by 2.2- to 3.1-fold [36,37]. The ability to cope with sudden changes in daily life is impaired due to cognitive and motor inflexibility [38,39] attributable to nigrostriatal dopamine depletion in individuals diagnosed with PD. A recent study during COVID-19 revealed higher perceived stress in patients with PD compared with healthy controls, notably more so in participants with PD who reported worsening motor symptoms compared with those without such worsening [40].

There is a non-motor phase prior to the onset of the well-known motor symptoms in PD, with non-motor issues constituting the initial presentation in 21% of pathologically proven PD cases [41]. Numerous studies have highlighted the characterization of PD by an extended preclinical, non-motor phase, often featuring symptoms of depression [42,43,44,45]. Even before formal diagnosis, patients with PD can develop depression and anxiety, frequently occurring in the context of chronic stress or past emotional trauma [13,46,47]. These psycho-physiological states can influence the brain’s structure progressively through the HPA axis [13], causing abnormal hormonal secretion and worsening cognitive function.

A recent longitudinal study following a cohort of patients with PD over 7 years demonstrated that individuals with comorbid anxiety and/or depression experienced significantly poorer outcomes than their non-depressed or non-anxious counterparts [48]. These patients had lower Schwab and England Activities of Daily Living (SE-ADL) scores and higher total scores on the MDS-UPDRS. The initiation of treatment for depression was associated with improvement in SE-ADL scores and reductions in MDS-UPDRS total scores over time, while treatment for anxiety correlated with lower total levodopa equivalent daily doses (LEDD) over the follow-up period. Most patients received SSRIs, with other prescribed SNRIs, TCAs, benzodiazepines, or atypical antidepressants. Additionally, a meta-analysis involving over 2000 patients with PD suggested that early treatment with tricyclics, particularly amitriptyline, was associated with a delay in the initiation of dopaminergic therapy, and that antidepressant treatment may induce neuroprotective changes that reduce the rate of dopaminergic cell degeneration in PD [49].

The proposed physiological pathway by which psychological stress affects the progression and symptoms of PD has been described in several reports [50]. Dopamine, a catecholamine modulatory neurotransmitter with both inhibitory and excitatory functions [51], plays a crucial role. Stress was demonstrated to disrupt dopaminergic projections in both the mesocortical and mesolimbic systems [12,52]. These dopaminergic pathways are integral to the reward system, and the effects of chronic stress on reward perception leading to depression are attributed to interactions between the dopaminergic system and the HPA axis, as well as between the dopaminergic system and the serotonergic system [53,54]. Several reports have demonstrated that early psychological stress that activates the HPA axis exacerbates dopamine depletion and correlates with a reduction in dopamine synthesis in the brain [12,13,55]. Consequently, a deficiency in dopamine resulting from early life stress may, in certain instances, predispose an individual to depression and eventually neurodegenerative pathologies, such as PD. Elevated stress levels can further reduce the effectiveness of levodopa treatment, leading to an additional decline in motor symptoms [56]. Furthermore, a recent study conducted with transgenic mice overexpressing mutant α-synuclein demonstrated that psychological stress can increase the susceptibility of PD through the release of corticosteroids, upregulation of *α*-synuclein, and facilitating membrane phospholipid peroxidation. This process led to lipid peroxidation in the plasma membrane, and ferroptosis of the dopaminergic neurons [57].

## 4. Stress and Inflammation

Chronic inflammation is a persistent, low-grade, and systemic condition associated with various chronic diseases, including T2DM, as well as metabolic, cardiometabolic, and psychiatric disorders [58]. Stress is another factor contributing to this complex relationship. The balance between the generation and removal of ROS and their derivatives is disrupted, leading to the onset of oxidative stress in cases of chronic stress and inflammation [59]. Additionally, individuals experiencing prolonged stress often undergo changes in their dietary habits. An illustration of this intricate relationship is the reported observation that individuals who are obese and insulin-resistant are more susceptible to major depressive disorders (MDD) compared to their healthy counterparts [60,61].

Chronic stress has also been identified as a trigger for proinflammatory factors, including cytokines and chemokines, which subsequently activate the HPA axis [62] and contribute to nigral cell death in PD [63,64]. Cytokine upregulation has even been linked to sudden death following severe emotional trauma [65]. Dysfunction of inflammatory markers such as tumor necrosis factor (TNF)-α, interleukin (IL)-1β, IL-6, and IL-10, and transforming growth factor (TGF)-β, has been correlated with the exacerbation of PD symptoms [12,22,66]. TNF-α, Ils, and β2-microglobulin have also been observed in the substantia nigra of patients with PD [67]. In addition, dysregulation of the HPA axis can lead to dendritic remodeling, neurogenesis dysfunction, apoptosis in hippocampal neurons, and an increase in oxidative stress [68].

Stress was suggested to reduce regulatory T-lymphocytes by 50% in individuals with post-traumatic stress disorder [69], and a similar reduction has been observed in patients with PD as well [70]. It was also considered that the dysfunction of regulatory T-lymphocytes may contribute to the loss of dopaminergic cells [71].

There is an elevation in cytokine levels in the setting of physical and/or psychological stress. Individuals with MDD exhibit heightened levels of pro-inflammatory cytokines, notably TNF-alpha and IL-6. Both TNF-alpha and IL-6 hinder vascular vasodilation by inducing the expression of chemokines in endothelial cells and adhesion molecules, ultimately contributing to hypertension [72]. Increased levels of other chemokines and cellular adhesion molecules, such as human macrophage chemoattractant protein-1 and E-selectin, along with additional pro-inflammatory cytokines and acute phase proteins, such as C-reactive protein (CRP), alpha-1-acid glycoprotein, and haptoglobin, have been identified in patients with depression. These factors are regarded as risk factors for the development of metabolic syndrome [73].

Despite the anti-inflammatory nature of glucocorticoids in the short term, prolonged and chronic exposure to glucocorticoids, as observed in depression and chronic stress, results in the insensitivity of peripheral and central glucocorticoid receptors due to receptor internalization from the cell surface [74]. Additionally, the reduced activity of the insulin receptor glucocorticoids leads to the downregulation of GLUT4, causing decreased glucose transport into the brain [75]. This may lead to insulin resistance, which has been suggested to play a role in the pathophysiology of PD.

## 5. Stress and T2DM

The connection between both acute and chronic stress and T2DM is well-established. Individuals with DM are known to be more susceptible to anxiety, depression, anger, and other stress-related behaviors than those without DM [76,77]. However, this correlation can also work in the reverse direction. There is abundant evidence that stressful events, such as trauma, workplace stress, and emotional stress, can negatively impact glucose homeostasis and induce insulin resistance [78]. Stress can also compromise a healthy lifestyle by leading to overfeeding and hyperphagia, especially with high-caloric food, as well as promoting a sedentary lifestyle and low levels of physical activity—all of which further contribute to a state of insulin resistance [79,80,81]. For example, an animal study that involved the use of an inescapable foot shock as an acute stress provoker showed acute development of insulin and glucose intolerance, especially through hepatic insulin resistance [82]. A recently published review explored the association between MDD, considered a form of chronic stress, and its link to chronic inflammation and insulin resistance [83].

MDD and chronic stress are believed to be interconnected through the HPA axis [84]. Depression elevates cortisol and catecholamine levels by modifying the HPA axis, thereby counteracting the hypoglycemic effects of insulin and leading to insulin resistance.

Research findings indicate a correlation between heightened sensitivity to CRP levels and the prevalence of metabolic conditions, such as central obesity, insulin resistance, and hypertension [85,86]. Stress has been shown to have a significant negative impact on beta cell function and glucose homeostasis. Multiple studies have demonstrated that both acute and chronic psychological stress can promote beta cell death, impair insulin production and secretion, and lead to blood glucose fluctuations, particularly in individuals with or at risk for diabetes. The mechanism behind this relationship is likely related to stress-induced overstimulation of the immune response, resulting in increased levels of inflammatory mediators and pro-apoptotic agents in pancreatic islets, ultimately compromising beta cell function and potentially contributing to the development or progression of diabetes [87].

It was reported that chronic stress can lead to increased insulin resistance, making it harder for cells to absorb glucose from the bloodstream. Insulin not only performs its well-recognized metabolic functions but also influences cell growth, cognition, behavior, and neuroprotection [88]. Insulin can cross the blood–brain barrier via saturable receptor-mediated transport, which can be suppressed by obesity, prolonged peripheral hyperinsulinemia, or aging [89]. Insulin receptors are widespread throughout the brain even though brain cells do not require insulin for glucose uptake [90]. Importantly, postmortem studies on PD have shown evidence of impaired neuronal insulin signaling [91,92]. Insulin binds to insulin receptors and leads to the activation of the phosphatidylinositol 3-kinase (PI3K) signaling pathways, among others. PI3K triggers the activation of mammalian target of rapamycin and protein kinase-B, which regulate vital functions such as mitochondrial biogenesis, apoptosis, inflammation, and autophagy, all of which are crucial for the survival of the dopaminergic cell [93].

In addition to the PI3K signaling pathway, insulin can also regulate mitochondrial function via the peroxisome proliferator-activated receptor gamma coactivator-1-alpha pathway [94]. Insulin resistance and T2DM lead to low Parkin and PGC1α levels, which impair mitochondrial proteins and genome expression, downregulate polo-like kinase-2(PLK2), increase ROS production, and cause abnormal mitochondrial metabolism [95]. Mitochondrial respiration is impaired by increased intraneuronal glucose concentrations, leading to ROS formation, cell damage, oxidative stress, and ultimately neuroinflammation and neuron death [96]. Neurons with increased energy requirements, such as dopaminergic neurons, are more susceptible to impaired mitochondrial metabolism, making them more vulnerable to hyperglycemia [97]. This increased susceptibility of nigral neuronal cells to damage by diabetes has been described in animal [98] and in vitro models [99]. Furthermore, the role of insulin in neuroinflammation is not limited to neuronal cells, as the PI3K/Akt pathway has been reported to induce several anti-inflammatory signaling pathways, reduce oxidative stress, and facilitate cell survival and neuroprotection [100]. Insulin also regulates vesicular monoamine transporter 2, which controls dopamine toxicity.

Several enzymes have been shown to have a link between insulin resistance, DM, and neurodegeneration, amylin being one of the latter [94]. Also known as islet amyloid polypeptide, amylin is released by pancreatic beta cells in response to blood glucose and insulin. The amylin receptor serves several physiological functions, among them the regulation of dopamine signaling [101], and it is thought to have neuroprotective properties in several animal models [94,102].

Another enzyme is α-synuclein, which is phosphorylated at its Ser129 residue [103] by PLK2. As discussed before, the enzymatic activity of PLK2 is intricately regulated by excessive ROS levels and intracellular oxidative stress, promoting increased expression of PLK2 at both the mRNA and protein levels [104,105]. This process promotes the abnormal folding of α-synuclein, interfering with its degradation, and ultimately activates and enhances the mitochondrial and nuclear accumulation of α-synuclein. A recent study demonstrated that chronic stress (by corticosterone administration) enhances α-synuclein pathological changes, overrides compensatory mechanisms, and leads to more overt neurodegeneration and subsequent emergence of PD phenotypes [106]. The abnormal spreading of pathologic α-synuclein is generally considered to be linked to disease propagation in PD, similar to the abnormal folding of amylin. Furthermore, there appears to be a potential interaction between these two proteins in triggering and exacerbating the pathology in PD and T2DM. Amylin and α-synuclein were found to be abundant in the pancreatic β cells of patients with synucleinopathies, possibly supporting the occurrence of insulin resistance in PD and other neurological pathologies in the absence of T2DM [107,108]. Furthermore, amylin can interact with α-synuclein and accelerate its aggregation in vitro, providing a theoretical rationale for T2DM being considered a risk factor for PD [109]. This is in line with other research showing significantly increased levels of amylin and pathogenic α-synuclein in the substantia nigra of patients with PD compared with healthy controls [107,110]. Increased deposits have also been observed in cellular [111] and animal models [112], as well as in pathological studies in humans [107].

Several studies have investigated the interactions between T2DM and PD. A meta-analysis, published in 2011, concluded that in prospective studies, T2DM constitutes a risk factor for PD [113]. A subsequent 2016 meta-analysis, encompassing over 1.7 million individuals, found that the risk for PD in diabetic patients was enhanced by approximately 38% [114]. Moreover, substantial evidence indicates common biological mechanisms [115] and genetic links [116] between these diseases, emphasizing dysfunctional insulin signaling as a potential convergent pathway [117]. For example, studies have shown that patients with T2DM who do not have PD showed signs of subclinical striatal dopaminergic dysfunction on DaTscans [118]. Similarly, healthy mice subjected to a high-fat diet to induce peripheral insulin resistance demonstrate nigrostriatal dopaminergic dysfunction and parkinsonism [119]. This supports the notion that T2DM and PD are likely synergistic conditions linked by dysregulated pathophysiological pathways, rather than two coincidental aging processes.

Beyond its effects on PD risk, several studies have evaluated the influence of T2DM on PD progression, suggesting that comorbid T2DM may be associated with more severe motor [118,120] and nonmotor symptoms [121] as well as an increased risk of developing mild cognitive impairment (MCI) with faster cognitive decline and gait impairment [121].

It is intriguing to note that various diabetic medications have demonstrated protective properties in the treatment of PD. Examples of such medications include glucagon-like peptide-1 (GLP1) like exenatide [94]. A meta-analysis assessing the impact of exenatide in human trials for PD concluded that exenatide administration yielded benefits at 12 months across motor and cognitive scales, nonmotor issues, and even on the UPDRS IV scale [122]. Dipeptidyl peptidase 4 inhibitors (DPP4i), through the inhibition of GLP1 degradation, were shown to decrease the prevalence of PD in human patients, as indicated by case-control studies [123] and longitudinal cohort studies [124]. On the other hand, metformin’s role in PD remains a subject of controversy.

## 6. Concluding Remarks

Chronic stress can have far-reaching effects on molecular and cellular signaling pathways, accelerating the degeneration of dopaminergic cells and complicating the progression and severity of PD. The relationship between stress and insulin resistance and T2DM is very strong, and the two may be mutually dependent. This interaction manifests at the biochemical and molecular levels and leads to a variety of clinical manifestations of neurodegenerative diseases associated with various comorbidities. Understanding this linkage is crucial for developing DMTs and lays the foundation for personalized medicine, warranting further exploration.

## Figures and Tables

**Figure 1 ijms-25-12409-f001:**
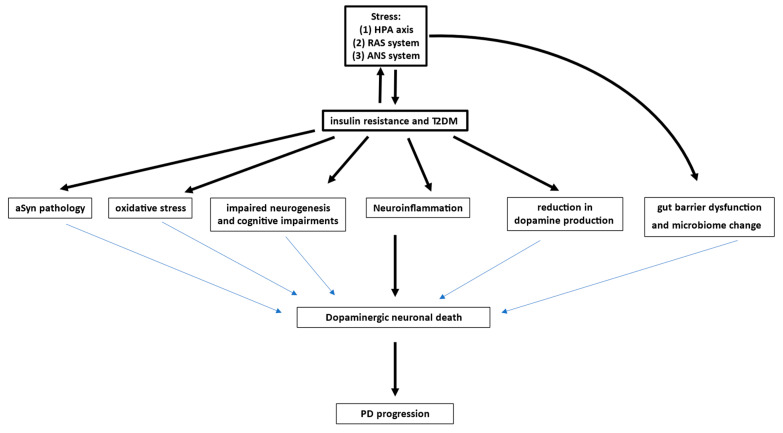
Hypothetical role of stress as a trigger for metabolic changes and dopaminergic neurodegeneration in Parkinson’s disease (PD). Potential link between insulin signaling and stress in the progression of PD.

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
