# Peer review of "The Interplay of Stress, Inflammation, and Metabolic Factors in the Course of Parkinson’s Disease"

_ijms, 2024, doi:10.3390/ijms252212409_

Round 1

Reviewer 1 Report

Comments and Suggestions for Authors

The authors aim to summarize interactions between emotional stress and selected key players in metabolism, such as insulin, as potential mechanisms underlying neurodegeneration in PD. This Review article is relative short and contains limited novel information.

My comments are as follows:

1.    The authors discuss PD in the Introduction section, but discussing mostly stress in the main text. Please be consistent in the focus.

2.    There should be a sub-section dealing with the interaction between DM and PD.

3.    This article can be benefited from adding some novel information between stress and PD, such as microbiota, BBB impairment, or epigenetic modifications, etc...

4.    Section 3 regarding stress and animal model does not fit well with the main text and is relatively superficial.

5.    Figures 1 should have the element of DM.

6.    A few typos or editing errors: inescapable (line 167);

Author Response

We are grateful to the reviewers for their thoughtful comments, which have helped us improve the paper. We have revised the manuscript in line with their suggestions. Below is our point-by-point response to each of the reviewers' notes, and a revised manuscript with the changes according to the reviewers' notes highlighted in yellow.

  1.    The authors discuss PD in the Introduction section but discussing mostly stress in the main text. Please be consistent in the focus.
    Answer: We added a paragraph in the introduction section regarding stress.
  2. There should be a sub-section dealing with the interaction between DM and PD.
    Answer: Done. Lines 274-298.
  3. This article can be benefited from adding some novel information between stress and PD, such as microbiota, BBB impairment, or epigenetic modifications, etc...
    Answer: The article discusses the connection between stress and Parkinson's throughout, at both the cellular and molecular levels. However, we also added an additional paragraph about the link between Parkinson's and psychological stress (in the section about psychological stress), as well as the connection between the GI system and Parkinson's (in the section of Physiological Stress and PD).

  4. Section 3 regarding stress and animal model does not fit well with the main text and is relatively superficial.
    Answer: Thank you for this note, animal studies are beyond the scope of this review and the paragraph was deleted.

  5. Figures 1 should have the element of DM.
    Answer: Done.

  6. A few typos or editing errors: inescapable (line 167);
    Answer: Done, thank you.

Reviewer 2 Report

Comments and Suggestions for Authors

In this review manuscript, the authors summarize the current understandings of stress conditions such as psychological stress and metabolic stress and Parkinson disease (PD). The topic is interesting and the manuscript is well-organized. The manuscript may be accepted after addressing my several concerns as follows.

1. Fig. 1, please delete “Graph 1 Stress … hypothesis”. I have several concerns regarding Fig. 1. Fig. 1 says all stress conditions induce insulin resistance, which eventually leads to PD progression. Is this correct? If so, it may helpful that the authors add a section for explain how stress conditions cause insulin resistance.

2. Regarding Fig. 1, please describe more details of stress conditions and impaired neurogenesis. Also, according to Fig. 1, reduction in dopamine production leads to dopaminergic neuron death. Is this correct? Please describe in more detail by citing appropriating references.

3. The contents of lines 67-74 seems to be one of the main topics of this review. Please cite appropriate original articles, not reviews.

4. Lines 75-83, regarding the animal models, please describe the animal models in more detail, i.e., rodents overexpressing mutant alpha-synuclein or MPTP-induced model etc.

5. Lines 173-174 needs a reference.

6. The major pathological hallmark of PD is the formation of Lewy bodies. Can the authors add some descriptions about alpha-synucleinopathy and stress conditions?

Author Response

We are grateful to the reviewers for their thoughtful comments, which have helped us improve the paper. We have revised the manuscript in line with their suggestions. Below is our point-by-point response to each of the reviewers' notes, and a revised manuscript with the changes according to the reviewers' notes highlighted in yellow.

  1. Fig. 1, please delete “Graph 1 Stress … hypothesis”. I have several concerns regarding Fig. 1. Fig. 1 says all stress conditions induce insulin resistance, which eventually leads to PD progression. Is this correct? If so, it may helpful that the authors add a section for explain how stress conditions cause insulin resistance.
    Answer: Fig. 1 reflects hypothetical role of stress as a trigger for metabolic changes and dopaminergic neurodegeneration in Parkinson’s disease and potential link between stress conditions and insulin resistance, which in turn could result in PD progression, as described throughout the article. At no point is it claimed that these conditions necessarily lead to this outcome. There is a section in the article (Section 5) that elaborates on the connection between stress, insulin resistance, and type 2 diabetes.
    The figure's caption has been changed to reflect that this is a hypothetical possibility, not a certainty.: “Hypothetical Role of Stress as a Trigger for Metabolic Changes and Dopaminergic Neurodegeneration in Parkinson’s Disease (PD). Potential Link Between Insulin Signaling and Stress in the Progression of PD”.
  2. Regarding Fig. 1, please describe more details of stress conditions and impaired neurogenesis. Also, according to Fig. 1, reduction in dopamine production leads to dopaminergic neuron death. Is this correct? Please describe in more detail by citing appropriating references.
    Answer: Stress axes described in the article have been added to the graph, outlining the pathways through which stress leads to the various pathologies described in our article. The effects of stress on the GI system, and its contribution to Parkinsonian symptoms, as described in the article, have also been included. Cognitive impairment was added to impaired neurogenesis, based on the article from Hirsch EC et al… 2009, citation number 67 in our article.
     The articles from Moore H et al… 2001 and from Rasheed N et al… 2009, cited numbers 33 and 34 accordingly in our article, describe the decline in dopamine as leading to the destruction of dopaminergic cells. It is also described extensively throughout other section of the article, as shown in the section about psychological stress and highlighted in yellow.

  3. The contents of lines 67-74 seems to be one of the main topics of this review. Please cite appropriate original articles, not reviews.
    Answer: Thank you, Done. All the articles, except for the article from Jobes, ML… 2008, cited number 22 in our article, have been replaced with the original sources.
    Regarding Article 22, there is no single study or paper that demonstrates the quoted statement. It involves a broad range of articles that go beyond the scope of the current paper, making it impossible to cite just one study. Therefore, a review article is needed to represent this information.
    The new revised sections appear in lines 77-84 and highlighted in yellow, as well as the new references which are also highlighted.

  4. Lines 75-83, regarding the animal models, please describe the animal models in more detail, i.e., rodents overexpressing mutant alpha-synuclein or MPTP-induced model etc.
    Answer:  After reevaluating the review, we determined that animal models are beyond its scope and have therefore excluded them.

  5. Lines 173-174 needs a reference.
    Answer: Done

  6. The major pathological hallmark of PD is the formation of Lewy bodies. Can the authors add some descriptions about alpha-synucleinopathy and stress conditions?
    Answer: Done. We included a paragraph regarding alpha-synucleinopathy and stress conditions in the section about stress and T2DM.
    Thank you.

Reviewer 3 Report

Comments and Suggestions for Authors

The manuscript of Tal ben Shaul and co-workers summarizes the possible role and interplay of stress, inflammation and metabolic factors, especially insulin, in the pathomechanism of Parkinson’s disease (PD). The manuscript is of interest, however, there are some questions/remarks.

It seems convincing that stress, inflammation and insulin resistance contribute to the pathomechanism of PD. The authors state that “An illustration of this intricate relationship is the reported observation that individuals who are obese and insulin-resistant are more susceptible to major depressive disorders (MDD) compared to their healthy counterparts”. Are there data that individuals with insulin resistance or T2DM have a higher risk for PD? Is this specific for PD or other neurodegenerative disorders such as Alzheimer’s or Huntington’s disease as well? As far as DMTs are concerned, these treatments are supposed to be specific for PD or more general? Are there data related to possible beneficial effects of anti-depressants or T2DM medication in PD either alone or in combination of levodopa?

Author Response

We are grateful to the reviewers for their thoughtful comments, which have helped us improve the paper. We have revised the manuscript in line with their suggestions. Below is our point-by-point response to each of the reviewers' notes, and a revised manuscript with the changes according to the reviewers' notes highlighted in yellow.

It seems convincing that stress, inflammation and insulin resistance contribute to the pathomechanism of PD. The authors state that “An illustration of this intricate relationship is the reported observation that individuals who are obese and insulin-resistant are more susceptible to major depressive disorders (MDD) compared to their healthy counterparts”. Are there data that individuals with insulin resistance or T2DM have a higher risk for PD? Is this specific for PD or other neurodegenerative disorders such as Alzheimer’s or Huntington’s disease as well? As far as DMTs are concerned, these treatments are supposed to be specific for PD or more general? Are there data related to possible beneficial effects of anti-depressants or T2DM medication in PD either alone or in combination of levodopa?

Answer:
The connection between insulin resistance, diabetes, and Parkinson's is extensively described in this article in section 5, stress and T2DM.
We have added a paragraph describing the relationship between diabetes medications and the treatment of Parkinson's at the end of section 5. Additionally, we added a paragraph describing the relationship between anti-depressants and PD in section 3. Several articles describe depression and anxiety as important factors in the prognosis of PD, and that adequate treatment led to an improvement in a number of diagnostic scores, as well as a delay in the initiation of levodopa treatment.

Round 2

Reviewer 1 Report

Comments and Suggestions for Authors

The authors have addressed all the concerns raised by me.